# Structure and function of Full-length Tau

**Laura Vallés-Saiz**[1], **Indalo Domene-Serrano** [1,2], **Angel J. Picher**[3], **Mar Pérez**[4], **Vega García-Escudero** [1,4,5], **Félix Hernández** [1], **Jesús Avila** [1,5]*

**1** Centro de Biología Molecular Severo Ochoa, CSIC-UAM), Madrid, Spain, **2** Facultad de Ciencias Experimentales, Universidad Francisco de Vitoria, Pozuelo de Alarcon, Madrid, Spain, **3** 4basebio S.L.U., Parque Científico de Madrid, Cantoblanco, Madrid, Spain, **4** Anatomy, Histology and Neuroscience Department, School of Medicine, Universidad Autónoma de Madrid (UAM), Madrid, Spain, **5** Networking Research Centre on Neurodegenerative Diseases (CIBERNED), Instituto de Salud Carlos III, Madrid, Spain

* javila@cbm.csic.es

## Abstract

Tau protein, encoded by the MAPT gene, is a microtubule-associated protein involved in the regulation of microtubule stability in neurons, contributing to cell shape maintenance and intracellular transport, among other functions. Tau is not found as a unique isoform; instead, different Tau isoforms of varying sizes are present in the brain, but a Full-length Tau isoform (Full Tau) containing all 16 exons has never been previously identified. This study has explored the structure and function of the Full Tau isoform, which includes all exons of the MAPT gene. To achieve this, we expressed the Full Tau isoform in bacteria, alongside the Tau 4R2N isoform as a control, and tested its microtubule-binding capacity, self-aggregation propensity, and effects on cultured cells regarding cell proliferation and cell death. Our results indicated several differences between the Full Tau and Tau 4R2N isoforms, suggesting distinct roles in cellular dynamics. To explain these differences, we suggest the role of exon 8, which is present in the Full Tau isoform but absent in Tau 4R2N.

## Introduction

Tau protein is one of the brain microtubule-associated proteins present in neuronal cells [1,2]. Tau is crucial for regulating microtubules stabilization [3], tubular structures that are part of the cytoskeleton and are essential for maintaining cell shape and for intracellular transport. In the brain, Tau regulates microtubule stabilization that are critical for the development and maintenance of neuronal connections [4–9]. The gene that encodes the Tau protein is MAPT gene, located on chromosome 17q21 [10]. Three main Tau transcripts derived from the MAPT gene have been characterized, each arising through regulated alternative splicing events [11]. Although alternative splicing of MAPT RNA results in several Tau isoforms, no transcript has yet

**Data availability statement:** Data is available in data repository: https://doi.org/10.6084/m9.figshare.28351508.v5.

**Funding:** This work was supported by grants from the Spanish Ministry of Science (MICIU): PID2023-149460NB-I00 (Dr. Felix Hernández), PID2021-123859OB-I00 (Dr. Jesús Avila), and PID2024-1554470OB-I00 (Dr. Jesús Avila), from MICIU/AEI/10.13039/501100011033/FEDER, UE (Financial) Laura Vallés-Saiz received a salary from the PID2021-123859OB-I00 project from MICIU. 4basebio S.L.U. Supported A.J.P. in the form of salary and research materials for the study. The Centro de Biología Molecular Severo Ochoa (CBMSO) is a Severo Ochoa Center of Excellence (MICIU, award CEX2021-001154-S). The funders had no role in study design, data collection and analysis, decision to publish, or preparation of the manuscript.

**Competing interests:** The authors have declared that no competing interests exist.

**Abbreviations:** AD, Alzheimer's disease; AF2, AlphaFold 2; CNS, central nervous system; MAPs, microtubule-associated proteins; MTs, Mouse brain microtubules; PNS, Peripheral Nervous System; pLDDT, predicted local distance difference test; NTR, N-Terminal Region; PRR, Proline-Rich Region; MTBR, Microtubule binding Region; CTR, C-Terminal Region.

been identified that translates into the Full-length Tau protein examined in this study. [8,11,12].

In the human central nervous system (CNS), six main distinct Tau isoforms are expressed, each contributing to functional diversity and complexity in neuronal operations [5,13,14]. These isoforms share a set of constitutive exons—1, 4, 5, 7, 9, 11, 12, and 13—that are consistently present across all variants, demonstrating their roles in Tau protein structure such as microtubule binding, nuclear acid binding or pathological self-aggregation [15]. Conversely, the non-constitutive exons 2, 3, and 10, are included in specific isoforms, reflecting their regulatory roles in isoform-specific functions [5,9,14]. Isoform diversity results from alternative splicing. a process in which specific exons are either included or excluded depending on the action of regulatory proteins that promote or inhibit exon inclusion [16,17]. The generation of diverse protein isoforms from a single gene via alternative splicing is not as common as one might believe [18]. RNA-seq studies indicate that almost all coding genes have a primary protein isoform, and only a few genes show strong evidence for multiple alternative splice isoforms, with MAPT being one of the well-studied examples [19–21].

Furthermore, recent studies have described new Tau isoforms obtained through intron retention rather than alternative splicing. These isoforms retain intron 3, 11, or 12 [22–24]. This observation suggests that exons 1, 12, and 13 may be considered non-constitutive exons. Despite the discovery of these new isoforms, none results in a transcript containing all 16 exons of the MAPT gene.

There are two types of Nervous System. First, CNS develops after the formation of the neural tube. Afterwards, cells from neuronal crest migrate to form the Peripheral Nervous System (PNS) [25]. Sometimes CNS and PNS are considered as different entities with transitions between both systems [26], such as in the case of dorsal root ganglion [27]. Additionally, specific Tau isoforms are present in mammal peripherical nervous system, where exons 4A and 6 are found as Big Tau isoform [28,29]. This big Tau isoform, may be protective for Tau misfolding, less vulnerable to Tau pathology and, as indicated, binds to microtubules from PNS [8]. The presence of exon 8 has been detected in cDNA derived from bovine RNA transcripts [30,31]. Also, exon trapping assays indicate that the exclusion of this exon is the default splicing pattern [9,12]. However, the inclusion of exon 8 in mature RNA transcripts has been observed not only in bovine but also in the monkey brain [32]. This pattern contrasts significantly with findings in humans, where exon 8 is typically not included in mature Tau isoforms, highlighting a notable difference in RNA processing between species [31].

Although the ENSEMBL database (MAPT-205; ENSP00000410838.2) indicates there is a transcript capable of being translated into Full Tau protein can exists, there are currently no reports of its expression. In this work, we will examine the properties, structure and functions of Full Tau (RNA transcript contains 16 exons: −1, 1, 2, 3, 4, 4A, 5, 6, 7, 8, 9, 10, 11, 12, 13 and 14, totaling 776 amino acids (known in this work as Full Tau; (Fig 1A). Exon −1 is part of the promoter and is transcribed but not translated, similar to exon 14) [33].

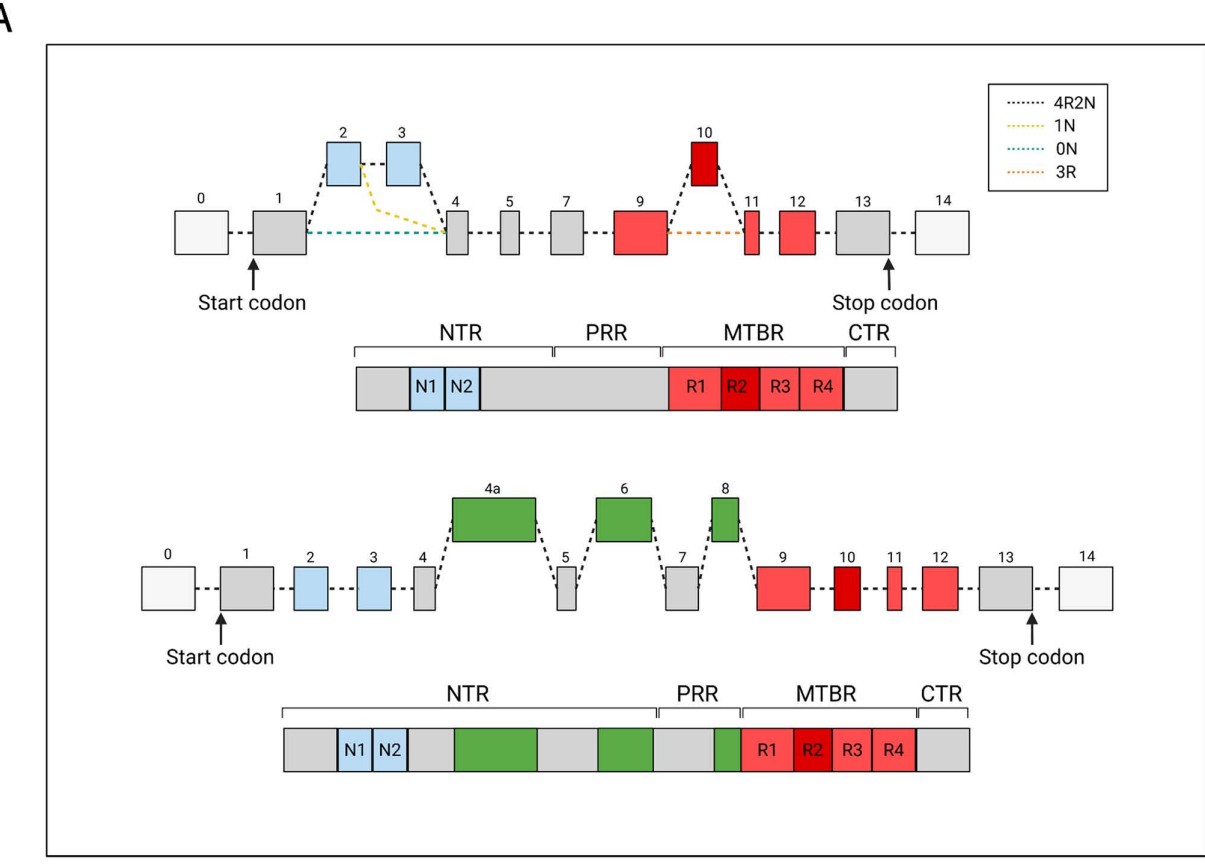

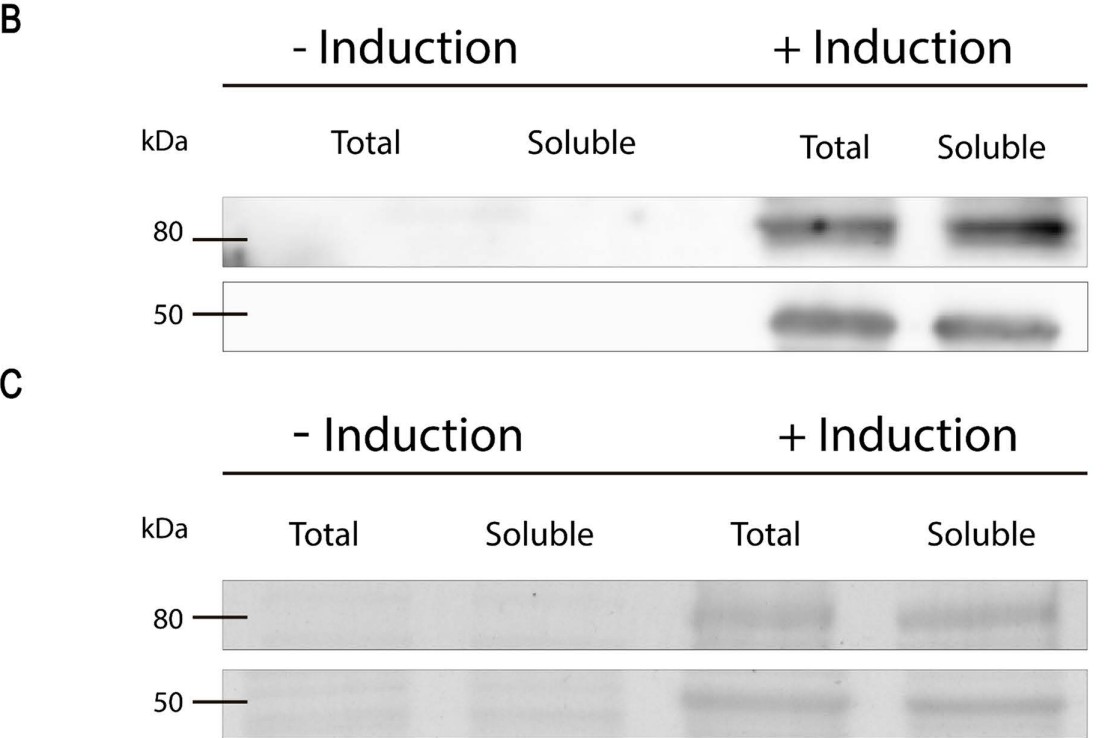

**Fig 1. Tau 4R2N and Full Tau protein purification. (A)** Alternative splicing scheme of Tau 4R2N and Full Tau and specific domains. NTR: N-Terminal Region, PRR: Proline-Rich Region, MTBR: Microtubule-Binding Region, CTR: C-Terminal Region **(B)** Coomassie brilliant blue staining of purification of

both proteins showing control of both expressions without and with induction of expression using 0.5mM IPTG. **(C)** Western Blot using the antibody that recognizes human Tau (HT7) indicating bands from induction are corresponding Tau. Both **B** and **C** panels are electrophoresis from same purifications. Unadjusted and uncropped images of blots and gels are found on S2 File.

## Results

### Isolation and purification of Full Tau protein isoform

As previously indicated, we have focused this work on the Tau protein isoform containing all the exons. We searched databases such as UniProt, NCBI, and ENSEMBL to look for the presence of Full Tau. Thus, Full Tau protein could be found in ENSEMBL under the identifier (MAPT-205; ENSP00000410838.2) but it seems that it could be present in humans and other mammals, in a very low amount, not only at protein but also at RNA level. Obviously, the translated protein is expected to be expressed at very low levels when its RNA expression is also low. Thus, to isolate Full Tau protein, we synthetized the cDNA for the RNA expressing all the exons and cloned that cDNA in a plasmid to be expressed in bacteria. As control we did the same experiment to express human Tau 4R2N isoform [34].

Fig 1 shows Tau 4R2N molecule and Full Tau molecule (Fig 1A), these proteins were compared to identify structural differences. The complete amino acid sequences of both human Tau isoforms are shown in S1 File. Tau 4R2N includes exons +2, +3, and +10 (S1A File) while Full Tau additionally incorporates exons +4A, +6, and +8, with a total of 776 aminoacids (S1B File). In both sequences, the exons are highlighted alternating between blue and black to facilitate identification.

As indicated, we induced the expression of both proteins in *E. coli* BL21 strains for subsequent *in vitro* analyses. The isolation of isoforms is well described in Materials & Methods. Fig 1B shows the Coomassie-stained gel control of both expressions without and with induction of expression using 0.5mM IPTG. In section C, the Western Blot using the antibody that recognizes human Tau (HT7) confirms that the new bands are Tau. Both Coomassie-stained whole gel control and Western Blot images are in S2 File. The results highlight the efficiency of the inducible expression system used, ensuring that overexpression of the isoforms of interest occurs only under controlled conditions. The clear differentiation in the amino acid sequences between Tau 4R2N and Full Tau underscores the importance of studying these isoforms individually to understand their specific roles.

### Full Tau, microtubule binding capacity

The primary and most well-known function of the Tau protein is its binding to microtubules [5]. To confirm the *in silico* findings, we decided to study the microtubule-binding capacity of our Full Tau isoform by separating microtubule-bound and free Tau by *in vitro* polymerization, followed by electron microscopy analysis and by separating polymerized or aggregated from soluble protein through centrifugation [35].

To analyze the binding of Full human Tau to microtubules and compare it with the binding of the human Tau 4R2N isoform, we used the same microtubules. These microtubules (MTs) are derived from mouse brain, obtained after a first polymerization cycle (see scheme in Fig 2A). These mouse brain MTs already contain mouse Tau protein. After cold depolymerization, tubulin-Tau links remain during a second polymerization cycle [1]. Before this second cycle, equal amounts of Full Tau or Tau 4R2N isoforms were added to depolymerized mouse microtubules. The second cycle was then carried out, allowing the added human Tau isoforms to bind to the second-cycle polymerized MT at the open sites present in the pre-assembled MT [36].

After polymerization, the assembled MTs were visualized by electron microscopy (Fig 2B). In MTs in the absence of human Tau isoforms or in the presence of human Tau 4R2N, MTs are clearly formed. However, in the presence of human Full Tau some differences in the amount and structure of MT were observed. A lower amount of MTs was found compared with those found in the presence of Tau 4R2N and a higher proportion of amorphous aggregates were observed (Fig 2B).

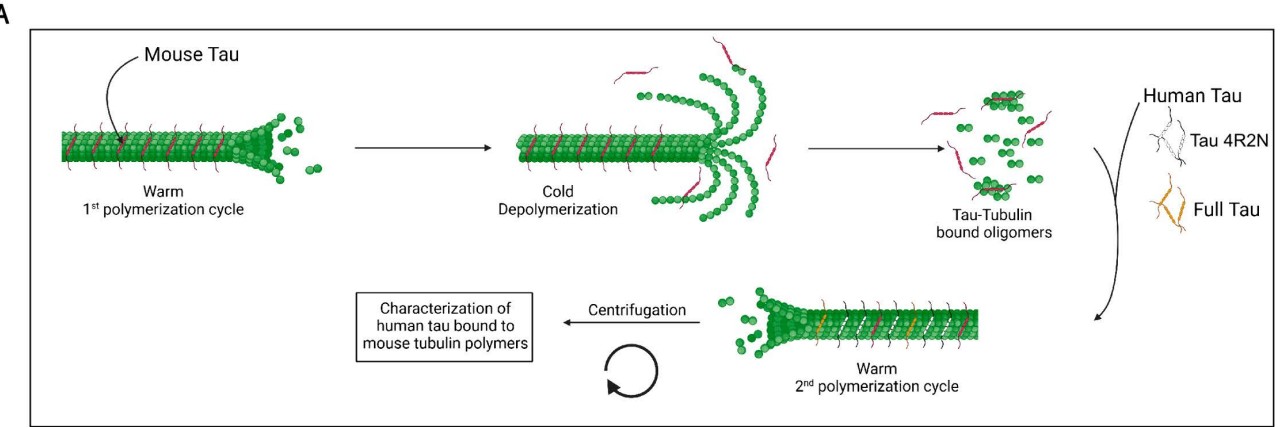

**Fig 2. Microtubule binding capacity of Full Tau isoform. (A) Scheme of the interaction of human Tau isoforms to second polymerization cycle of mouse microtubules (see Methods). (B)** Electron microscopy images of second polymerization cycle of mouse brain microtubules in the absence (control) or presence of Tau 4R2N or Full Tau isoform. Circles indicate thin MTs. **(C)** Coomassie brilliant blue staining showing tubulin percentage after centrifugation of the two cycle-polymerization MTs. SN: Supernatant, PLL: Pellet **(D)** Western Blot analyses of human Tau bound to microtubules after copolymerization, in the second polymerization cycle of mouse microtubules, detected by anti-human Tau antibody HT7. Unadjusted and uncropped

images of blots and gels are found on Data Availability. **(E)** Quantification of the ratio between Tau 4R2N and Full Tau Western Blot signal intensity in **C.** Quantitative analysis shows the mean ± SEM. * p < 0.05; ** p < 0.01 using one-way ANOVA; *post-hoc* Turkeys's test, followed by Student's t-test for comparisons. Sample size: n = 3.

Microtubule-like polymers with a smaller diameter than control microtubules were also found (Fig 2B). All these observations indicate that the presence of Full Tau isoform may partially interfere with the previously reported *in vitro* MTs polymerization taking place in the absence of that isoform.

Tubulin polymers in the absence or presence of human Tau isoforms, could be isolated by centrifugation and in the pelleted fraction, the presence of tubulin can be visualized by fractionation of the MTs proteins by gel electrophoresis, and staining with Coomassie blue (Fig 2C). By comparing the results of Fig 2B and 2C we can conclude that there is no major difference in the amount of pelleted protein, but there is a clear structural difference in the tubulin aggregates present in the pellet.

### How does the presence of exon 8 affect Tau binding to microtubules?

Figs 2D and 2E demonstrate that Full Tau exhibits reduced binding to microtubules compared to the Tau 4R2N isoform. Full Tau closely resembles Big Tau, an isoform containing all Tau exons except exon 8 [8]. However, previous studies [37] have shown that Big Tau binds to microtubules with higher affinity than that now found for Full Tau. To investigate the impact of exon 8 on Tau-microtubule binding, we examined the binding of an artificial isoform composed of Tau 4R2N plus exon 8. The results presented in S3 File reveal that this isoform has a significantly reduced microtubule-binding capacity, similar to that found for Full Tau isoform and distinct from that of Big Tau isoforms [32]. These findings suggest that the presence of exon 8 decreases the interaction of Tau with microtubules, explaining the weaker microtubule binding observed for Full Tau compared to the Tau 4R2N isoform.

### In vitro aggregation of Full Tau

We decided to study the aggregation capacity of the Full Tau isoform, as this is a crucial aspect in the research of neurodegenerative diseases such as Alzheimer's disease (AD) and other tauopathies [5,38]. Tau is a microtubule-associated protein that, under pathological conditions, can form insoluble aggregates, a central pathological feature in these tauopathies.

Fig 3A shows representative electron microscopy images illustrating the influence of heparin on the aggregation of Tau 4R2N and Full Tau isoforms, as it is well established that the presence of heparin accelerates filament polymerization for Tau [39,40]. In the presence of heparin, Tau 4R2N isoform, forms dense and highly organized structures (filaments). On the other hand, Full Tau in the presence of heparin, shows less aggregated structures that are more disorganized and amorphous compared to those assembled in the presence of Tau 4R2N (Fig 3B–BG). To test that those aggregates are indeed assembled from Full Tau, immuno electron microscopy was carried out by using an antibody against Tau. In S4 File, the arrow tips show the gold nanoparticles indicating Tau presence in those aggregates. In the absence of heparin, Tau 4R2N forms numerous amorphous aggregates and filaments, whereas Full Tau forms amorphous aggregates, though with fewer polymers (S5 File).

To quantify the amount of Tau aggregates, following heparin addition and incubation, the samples were centrifuged, and the presence of human Tau isoforms in the supernatant and pellet was determined. Fig 3F shows a larger amount of pelleted protein for the Tau 4R2N isoform compared to the human Full Tau isoform, as quantified in (Fig 3H–BJ).

### In silico 3D structural analysis of human Tau protein isoforms: Tau 4R2N and Full Tau

To try to explain the previous results, we decided to perform a three-dimensional *in silico* analysis of both isoforms using the models with the highest confidence (i.e., highest predicted local distance difference test (pLDDT)) among the five

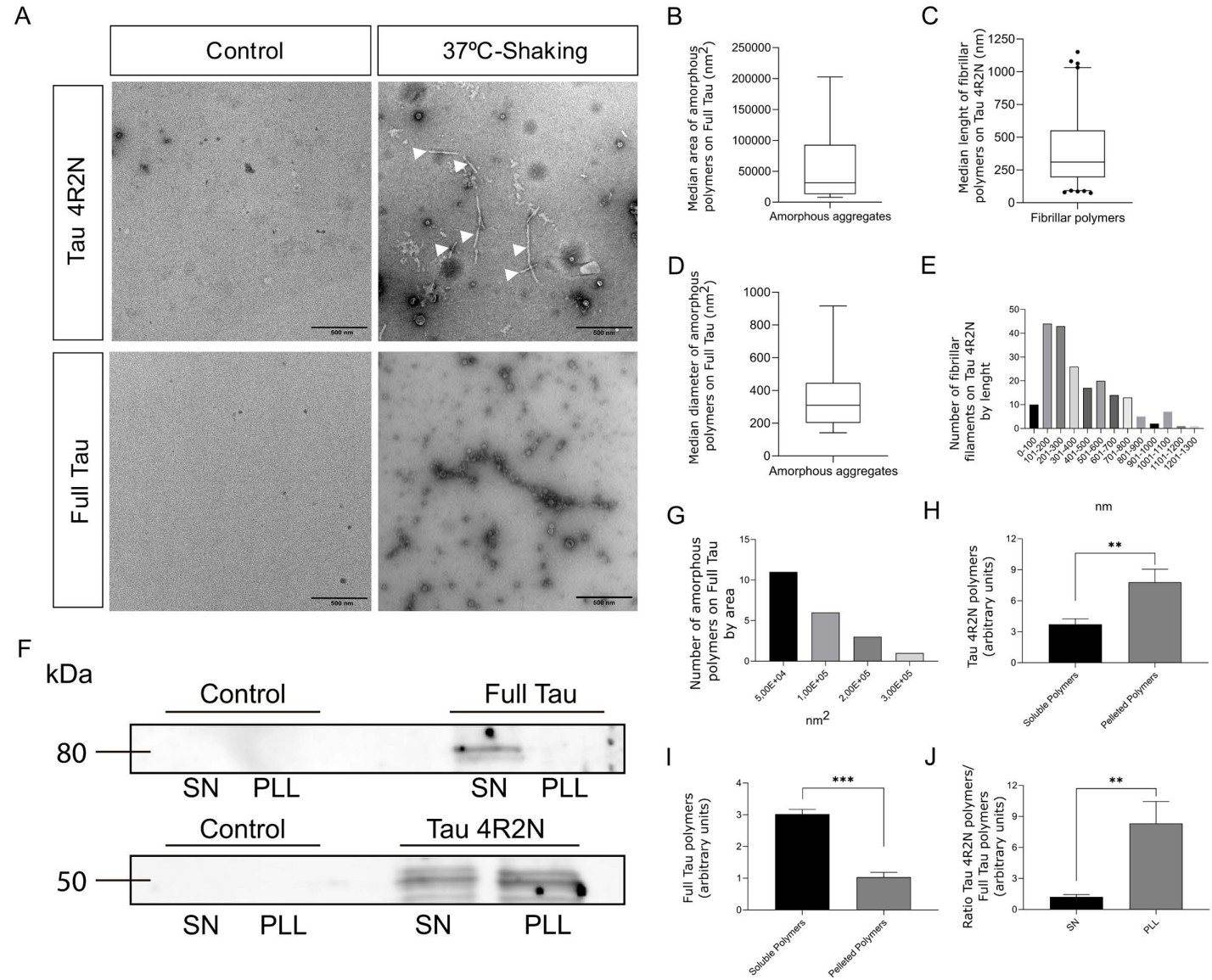

**Fig 3. Aggregation capacity of T42 and Full Tau isoforms. (A)** Representative images of electron microscopy of Tau aggregates obtained upon *in vitro* incubation of purified T42 and Full Tau extracts from bacteria in presence of heparin to promote aggregate formation. Filaments marked with white arrows ends. Scale bars show 400nm. **(B)** Area of the amorphous polymers found on Full Tau *in vitro* aggregation analysis. **(C)** Length distribution of the fibrillar polymers formed by Tau 4R2N. **(D)** Number of amorphous aggregates grouped by area intervals. **(E)** Number of fibrillar polymers of Tau 4R2N grouped by length intervals. **(F)** Western Blot of Tau isoforms present in soluble or pellet fractions after centrifugation. SN: Supernatant, PLL: Pellet Unadjusted and uncropped images of blots and gels are found on Data Availability. **(G)** Quantitative analysis of pelleted proteins. **(H)** Quantitative ratio of Tau 4R2Nand Full Tau protein. Quantitative analysis shows the mean ± SEM. * p < 0.05; ** p < 0.01 using unpaired; 2-tailed t test. Sample size: n = 3.

predictions generated by AlphaFold2 (AF2) [41]. Fig 4 shows the three-dimensional structure predictions of the Tau 4R2N and Full Tau isoforms, generated with AF2. The microtubule-binding domains (exons 9, 10, 11, and 12) are highlighted in red, exon 8 is highlighted in green, while the rest of the structure is shown in white. These predictions were visualized at different rotation angles (0°, 90°, 180°, and 270°) around the "y" axis to provide a comprehensive view of the conformation

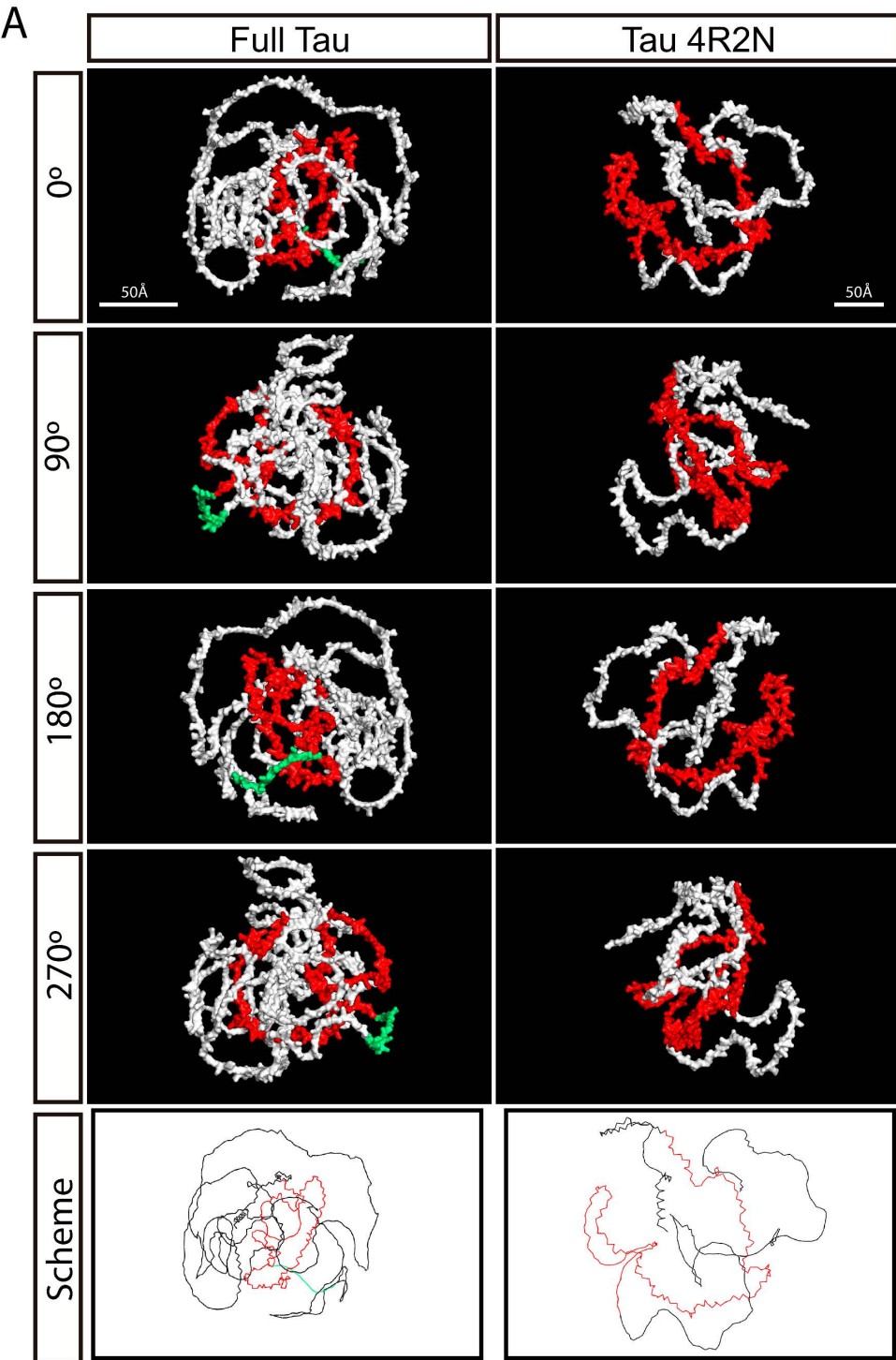

**Fig 4. Tridimensional structure prediction of Tau 4R2N and Full Tau.** Microtubule Binding Domain Exons' 9, 10, 11, 12 are represented in red, the rest of exons in white. Both are the predictions with the highest confidence (highest pLDDT) among the five predictions made by AF2. Images were taken at different degrees of rotation about the "y" axis for complete visualization. Exon 8 is highlighted in green. MTBD is colorized in red. Scheme of proteins structure highlighting differences in both structure and size.

of both isoforms. The images reveal the spatial arrangement of the microtubule-binding domains, which are crucial for the function of Tau in microtubule stabilization.

The quantitative analysis of solvent-accessible surface area [42] reveals that Full Tau and Tau 4R2N present practically the same buried area percentage (S1 Table), but three-dimensional structure predictions performed with AF2 showed that the microtubule-binding domains in Tau 4R2N are more accessible compared to Full Tau. In all rotations (0°, 90°, 180°, and 270°), the red-highlighted exons of Tau 4R2N are more prominently exposed and appear less obstructed by other regions of the protein. This suggests that in Tau 4R2N, these domains can more easily interact with microtubules, facilitating a greater binding and stabilization capacity. However, in the Full Tau isoform, these domains seem to be more enveloped or partially covered, mainly exon 8 (Fig 4, green). This reduced accessibility could affect its ability to efficiently interact with microtubules.

## Differential Effects of Tau Isoforms on Cell Viability and Growth Rate in HEK 293T Cells

Neurons exhibit a significantly lower proliferation capacity compared to non-neuronal cells. One possible explanation is that cell proliferation requires dynamic microtubules to facilitate the formation of the mitotic spindle. However, in neurons, microtubules are highly stable and lack such dynamism; interphase microtubules are not depolymerized for subsequent assembly into the mitotic spindle, due to the presence of microtubule-associated proteins (MAPs), such as Tau [43]. Additionally, the stability of neuronal microtubules supports the development of their unique asymmetric morphology, as opposed to the more rounded morphology typically seen in non-neuronal cells (see reference [5]). Interestingly, when Tau is expressed in non-neuronal cells, some neuronal-like features can be observed [44]. To investigate this further, we expressed Full Tau and the Tau 4R2N isoform in a non-neuronal cell line to assess potential changes in cell proliferation.

The representative images in Fig 5A show the cellular proliferation of HEK 293T cells after infection with lentiviruses expressing Tau 4R2N and Full Tau, and its growth rate, quantitatively represented as the percentage of newly occupied area by the cells (Fig 5B). We decided to study cellular proliferation rate as it allows us to identify whether the overexpression of one of the isoforms could generate stress or lead to apoptosis due to excessive microtubule stabilization, which affects their ability to properly adhere, mitotic capacity and maintain their normal extended morphology. It has been previously described that this occurs with the overexpression Tau [44].

In the negative control condition, cells maintain a common growth rate for the cell line. However, cells upon lentiviral infection show proliferative changes. When we quantified these growth rate changes (Fig 5B), we found significant differences when comparing Tau 4R2N with negative control and Full Tau. However, no significant differences were found between negative control and Full Tau. This could indicate that the Full Tau isoform does not induce microtubule over-stabilization, as occurs with the Tau 4R2N
isoform.

Cell viability was also analyzed using calcein (Fig 5C-F). Calcein is a hydrophobic, non-fluorescent compound that easily penetrates living cells. Once inside, intracellular esterases hydrolyze calcein, converting it into a hydrophilic, highly fluorescent molecule that remains well-retained within the cytoplasm. First, negative control was compared with empty lentiviral transduction controls, and it was observed that infection caused a significant cell death (Fig 5C-D). Once the effect of cells transduction was determined, the cell viability of cells overexpressing Tau 4R2N and Full Tau isoforms was compared. The overexpression of Full Tau isoform did not lead to a reduction in cell viability compared to its infection control (C.Empty-Full Tau) (Fig 5E). In contrast, the overexpression of the Tau 4R2N isoform resulted in a statistically significant increase in cell death compared to its infection control (C.Empty-Tau 4R2N), the negative control, and Full Tau (Fig 5E-F). These results suggest that the expression of Tau 4R2N has a greater impact on altering cellular physiology than Full Tau. This is likely due to excessive microtubule stabilization by Tau 4R2N, which interferes with the normal dynamics necessary for cell division and cellular transport. Consequently, we observe distinct differences in how these isoforms

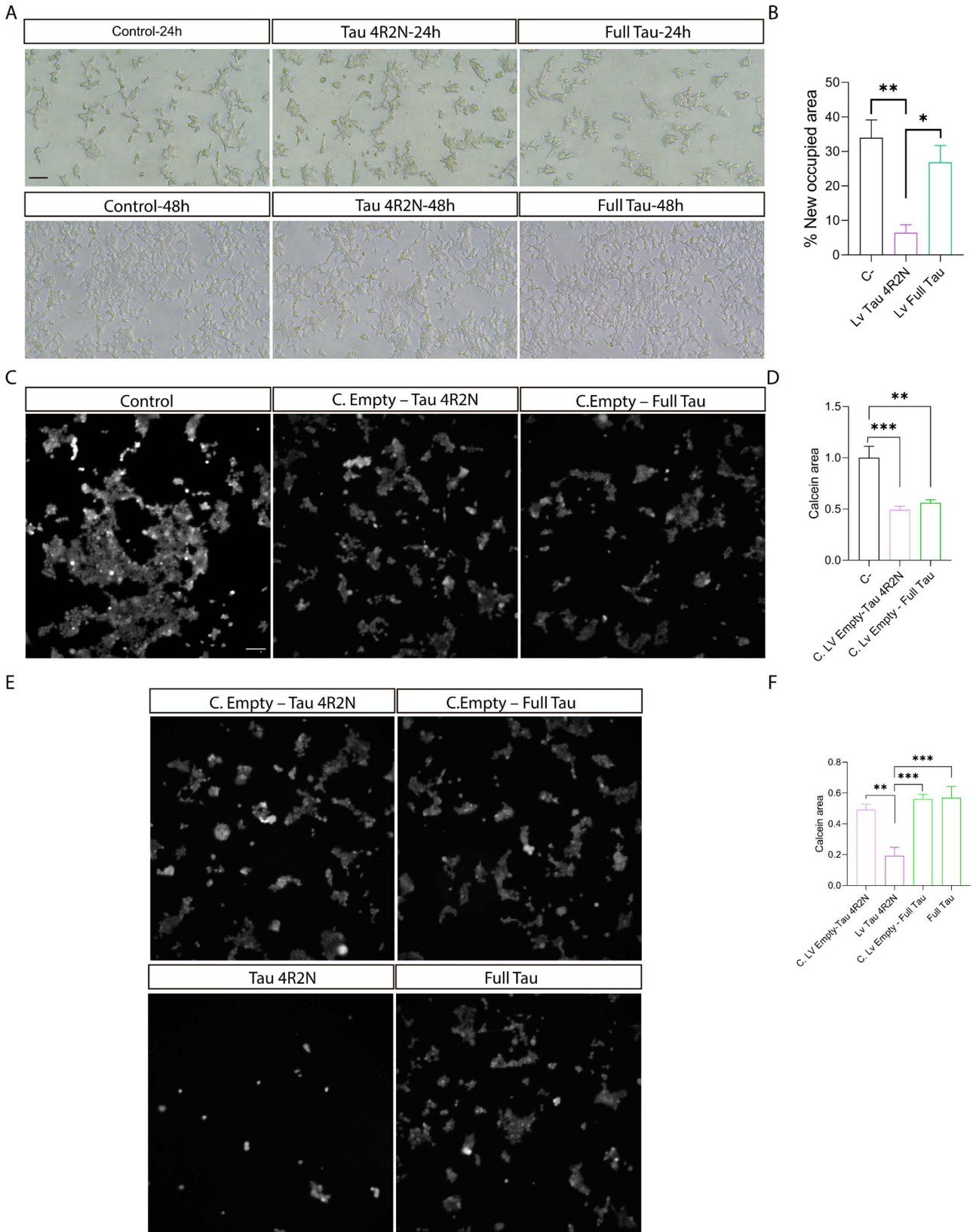

**Fig 5. Proliferation and viability of HEK 293T cells following lentiviral infection with Tau 4R2N and Full Tau.** **(A)** Cellular proliferation rate, assessed through images taken 24 hours post-transduction and after 48 hours of growth. Scale bar: 40 µm. **(B)** Quantification of cellular proliferation

based on the newly occupied area by cells. Sample size: n = 3 fields per replicate (n = 3). **(C)** Calcein viability assay comparing cell death between the negative control and transduction controls for each protein. Scale bar: 100 µm **(D)** Quantitative analysis of viability, measure from the data shown in (C). Sample size: n = 3 fields per replicate (n = 3). **(E)** Representative images of the calcein viability assay in cells overexpressing Tau 4R2N and Full Tau, along with their respective transduction controls (C. Empty-Tau 4R2N and C. Empty-Full Tau). Same scale as (E). **(F)** Quantification of cellular viability the data shown in (E). Sample size: n = 3 fields per replicate (n = 3). Quantitative data are presented as mean ± SEM. Statistical significance: *p < 0.05; **p < 0.01, determined by one-way ANOVA followed by Tukey's *post-hoc* test, with additional comparisons conducted using Student's t-test. Sample size: n = 5 fields per replicate (n = 3).

affect cellular structure viability and proliferation. These findings highlight the specific role of Tau 4R2N in disrupting cellular morphology, suggesting its potential impact on cellular functions related to microtubule dynamics.

## Discussion

Isolation of microtubules from mammalian brains results in the presence of not only the main component of microtubules, tubulin, but also microtubule-associated proteins (MAPs) such as MAP1A, MAP1B, MAP2, Tau, and others [2,45,46]. In all cases, except for the Tau protein, the Full-length protein is expressed, despite the presence of shorter isoforms [2,45–47]. However, there are currently no reports indicating the presence of the Full-length Tau isoform in the mammalian brain.

Curiously, shorter Tau isoforms are expressed in embryonic or first developmental stages, and larger isoforms are expressed at later times of development and maturation. Even in aged brains, the complete protein has not been detected, probably because the level of its RNA is very low, even at mature stages.

We decided to study a Tau isoform with all exons present in the MAPT gene, identified in ENSEMBL database as MAPT-205; ENSP00000410838.2, which we refer to as Full Tau in this work, and compare it with the well-described Tau 4R2N isoform [34]. A characteristic of the Full Tau isoform is that it includes all exons, which means the presence of exon 4A and exon 8 simultaneously. However, it has been described that both exons do not appear together in any known Tau isoform [8,14], highlighting the potential uniqueness of the Full Tau isoform.

We induced the expression of both isoforms in *E. coli* BL21 strains and conducted various *in vitro* assays with the proteins generated in our laboratory. Given that the microtubule-binding region [48,49] of Tau is crucial for its primary function and aggregation, and our *in-silico* analysis showed it to be less exposed in Full Tau, we performed experiments to evaluate the microtubule-binding capacity of our Full Tau isoform. Electron microscopy showed that, in the absence of added human Tau, the microtubules were well-formed, but their structure was denser and more intertwined in the presence of Tau 4R2N, suggesting a slight promotion of polymerization. Full Tau also slightly promoted polymerization.

Therefore, some differences on the interaction of Full Tau with tubulin to promote MT polymerization can be suggested. Firstly, the inclusion of these additional exons might cause Full Tau to adopt a conformation that partially blocks the microtubule-binding domains, particularly exon 8, as previously suggested. Although these findings are intriguing, more in-depth research is needed to fully understand their implications. Indeed, exon 8 sequence has some similarities with Exon 10 sequence (S6 File), and both can compete for the same microtubule polymer.

In both sequences, basic amino acids could react with the acidic residues present at the C-terminal region of tubulin molecules [50,51], but with different results: microtubule polymerization or aberrant aggregation (see Fig 2B), could take place in the absence or presence of Full Tau. In addition, Exon 8 could compete with some microtubules binding regions, like exon 10, in its interaction with tubulin monomers. Another possible explanation is that the more complex structure of Full Tau might favor intramolecular interactions that compete with microtubule binding. This means that some regions of the protein might interact with each other, preventing the microtubule-binding domains from being fully available for their stabilizing function [48,52,53]. More work should be done to determine the differences between Full Tau and other Tau isoforms with tubulin.

To study the aggregation capacity, we used heparin, which accelerates the polymerization of Tau filaments [4,54]. Electron microscopy showed that, in the presence of heparin, Tau 4R2N formed dense and organized structures, whereas

Full Tau formed less organized and more amorphous aggregates, with no filaments observed for Full Tau. Also, a lower amount of pelleted aggregates was found in Full Tau compared to Tau 4R2N isoform. This may be because self-aggregation occurs through the microtubule-binding regions of the Tau molecule, which could be partially obscured by the presence of exon 8, as previously indicated.

In this work, we conducted an *in silico* three-dimensional analysis with AF2 [41], revealing significant structural differences between the two isoforms, particularly in the accessibility or exposure of the microtubule-binding domains. In Tau 4R2N, these domains are more accessible compared to Full Tau, suggesting that Tau 4R2N may interact more effectively with microtubules. This could be due to potential interference in Full Tau by exon 8 with the microtubule-binding domain, as suggested by the AlphaFold model (Fig 4).

Finally, we analyzed the cell viability and growth rate of Hek-293T cells following infection with lentivirus expressing Tau 4R2N and Full Tau. Overexpression of Tau 4R2N significantly decreased the proportion of live cells compared to Full Tau overexpression. These changes suggest that Tau 4R2N, due to its excessive microtubule-stabilizing capacity, more drastically affects cell morphology, adhesion and proliferation than Full Tau, interfering with the normal dynamics necessary for cell division and transport.

A characteristic of mature neurons is the lack of proliferation due to cell cycle exit [55,56]. Neuronal microtubules are more stable than those in dividing cells [57] and are primarily used for forming complex structures related to their morphology [55]. Stable microtubules in neurons result from the binding of microtubule-associated proteins (MAPs), such as MAP1B, MAP2, or Tau, to brain tubulin, which could facilitate cell cycle exit and reduced cell proliferation. The action of any of these MAPs could complement the absence of other ones. Indeed, mice lacking Tau protein are viable although they have some defects [16].

Thus, to analyze the role of the Full Tau in the absence of other brain MAPs, or Tau isoforms, we have expressed Full Tau in non-neuronal cells, since in previous studies it was reported that other Tau isoforms expressed in non-neuronal cells facilitates microtubule stabilization and reduce cell proliferation [44]. Interestingly, Full Tau shows different effects, as indicated in this work.

In conclusion, the Full Tau isoform, possibly present in a low amount, shows less interaction capacity with microtubules and lower self-aggregation compared to the Tau 4R2N isoform. The structural characteristics of Full Tau affect the normal functionality of the Tau protein, suggesting that Full Tau could play a role distinct from that of the canonical isoforms and is likely to exert a non-deleterious influence on the pathological behavior of the Tau protein.

## Experimental procedures

### Bacteria culture and Tau purification

Vectors pRK-T42 and pRK-FullTau and pRK-T42 + E8 (GenScript, Netherlands), encoding different Tau isoforms were overexpressed in *E. coli* BL21 competent cells by electroporation, from which Tau was purified, as described elsewhere [54]. These bacteria were cultured in LB medium with 100 ng/ml of Ampicillin at 37 °C overnight. Bacterial suspensions were transferred to 1L of LB with 100 ng/ml of Ampicillin and further incubated at 37 °C, until the optical density readings at 600 nm ranged between 0.6 and 0.8. At this point, 0.5 mM of IPTG was added to each sample so as to trigger transcription and incubated once again at 30 °C for 3 h. The samples were centrifuged at 2950 × g at 4 °C for 20 min and pellets were resuspended in Tau buffer, which consists of Lysis buffer (20 mM MES pH 6.4, 1mM EGTA, 2 mM DTT) complemented with protease inhibitor mixture, 0.2mM $MgCl_2$, lysozyme and DNAse I. Subsequently, cells were disrupted with a French pressure cell press (in ice cold conditions to avoid protein degradation). In the next step, NaCl was added to a final concentration of 500mM and lysates were boiled for 20 min. Denatured proteins were removed by ultracentrifugation with 127.000 g at 4 °C for 30 min. To this supernatant, 2 mg/mL of streptomycin sulfate was added at 4°C for 15 minutes. The supernatants were kept to get rid of residual DNA. Ammonium sulfate $((NH_4)_2SO_4)$ (50–60%) was added to the samples,

and they were incubated agitating at 4 °C for at least 1 h. Subsequently, the Tau pellet was obtained by centrifugation at 15,000 g at 4°C for 30 minutes. The pellet was resuspended with buffer A (20mM MES pH 6.8, 1 mM EDTA, 2mM DTT, 0.1mM PMSF, 50mM NaCl) and dialyzed overnight at 4 °C against dialysis buffer A to remove salt. Next day, it was treated with RNAse (10 mg/mL). Pellets containing purified Tau were resuspended in buffer A and the purification process was checked by SDS-PAGE and Coomassie blue staining to check for the whole protein bands appeared upon Coomassie blue staining. The presence of human Tau isoforms was confirmed by Western blot using HT7 antibody (S2 File)

### Western blots

The proteins were separated on 10% acrylamide/bisacrylamide gels in the presence of SDS at 120 mV for approximately 1 h. The proteins in the gel were transferred to nitrocellulose membranes (Schleicher and Schuell, Keene, NH, USA) at 150 mA for 45 min, using the Bio-Rad Mini-Protean system. Subsequently, the membranes were blocked using 5% milk powder in 0.1% Tween PBS for 1 h. The membranes were then washed twice with 0.1% PBS/Tween-20 (v/v) under stirring for 10 min. The membrane was stained with Ponceau dye (Ponceau 0.3% in TCA 3%) to check the transfer efficiency. Finally, they were incubated with the appropriate primary antibody overnight at 4 °C: anti-human Tau (HT7, mouse, 1:5000, Thermo Scientific); anti-BetaTubulin (mouse, 1:5000, Sigma). Protein expression was detected using the secondary antibody (1:2000), which were incubated per one hour at room temperature. After performing three 10-min washes in the wash solution, the immunoreactive proteins were detected using the ECL (Enhanced Chemiluminescence Detection System, Amersham). Uncropped and unadjusted Western blot images are provided in S7 File.

### 3D modeling analysis

A user-friendly interface for accessing AlphaFold2 has recently been made available through notebooks. We used the ColabFold notebook, whose structure prediction is powered by AlphaFold2 combined with RoseTTAFold and a fast, multiple sequence alignment generation stage using MMseqs2. Each isoform structure prediction was downloaded from ColabFold and visualized with The PyMOL Molecular Graphics System, Version 2.0 Schrödinger, LLC. For accurate visualization, the microtubule binding domain region of both Tau isoforms was colored in red, while the rest of the protein was colored in white. Four images with different rotations along the y-axis were taken to provide a complete visualization of the protein structure.

### Tau microtubule-binding capacity assay

Tau microtubule polymerization was carried out as previously described [50,58], using the brain of 10-month-old C57BL/6 mice. After a first polymerization cycle, the microtubule fraction was collected by centrifugation. The microtubule pellet was resuspended in a buffer containing 0.1M MES pH 6.4, and a second polymerization cycle in the absence or presence of Full human Tau isoform or human Tau 4R2N isoform. After incubation for 30 min at 37 °C, electron microscopy [59], and the samples were centrifuged and the protein present in super and pellet fraction were characterized by gel electrophoresis and Western Blot using Tau antibody HT7 and quantified.

### Aggregation assays

Tau aggregates were grown for the different Tau isoforms by vapor diffusion in hanging drops, following the standard procedure employed for protein crystallization [60], see also [61]. The different Tau isoforms were purified from bacteria as outlined previously. Their concentration was estimated by measuring absorbance at 280 nm, factoring in individually determined extinction coefficients for each isoform, and confirmed via Coomassie Blue staining.

40 μM of Full Tau or Tau 4R2N isoforms in a buffer containing 0.1 M NaCl, 20 mM ammonium acetate ($CH_3COONH_4$) and 1.5 mM $CaCl_2$, pH 7.0 were incubated in the absence or presence (10 μM) of heparin for 12 h at 37 °C [61]. Protein

aggregates were analyzed by electron microscopy [54] and isolated by centrifugation [39], measuring the amount of protein in super and pellet by gel electrophoresis and Western Blot, using Tau antibody HT7, being quantified the obtained results.

## Immuno-MS

Post-incubation drops were adsorbed for 5 min onto 400 MEs Copper Collodion grids ionized in a BAE 120 Evaporator (Bal-Tec) and washed with PBS for 3 min. Grids were blocked for 10 min with PBS + 10% FBS and then incubated with Tau12 antibody (MAB2241, Sigma-Aldrich) diluted in PBS + 5% FBS for 30 min at RT. They were washed once in PBS for 3 min, then incubated with 10 nm colloidal gold conjugated with a secondary antibody diluted 1:30 in PBS and 5% fetal bovine serum (FBS) for 30 min. The grids were washed once in PBS for 3 min and once in H2O milli-Q for 3 min, then dried and stained with uranyl acetate for 40 s. The samples were visualized by transmission electron microscopy.

## Cell culture

Hek-293T cells (CRL-11268, ATCC) were cultured in DMEM supplemented with 10% fetal bovine serum, 2 mM glutamine, non-essential amino acids, 10 U/ml penicillin and 10 µg/ml streptomycin, at 37 °C and 5% $CO_2$.

## Lentivirus production

Viral stocks were produced in P100 plates by transient cotransfection of Hek-293T cells with 10 µg of the corresponding lentivector plasmid: Lv Tau 4R2N (kindly provided by Kenneth Kosik, UC Santa Barbara, California), Lv- Empty Tau 4R2N (GFP; Addgene plasmid 12252), Lv- Full Tau (GenScript, Netherlands) and Lv- Empty Full Tau (qWPI; Addgene plasmid #12254), 6 µg of the packaging plasmid pCMVdR8.74 (Addgene plasmid 22036) and 5 µg of the VSV-G envelope protein plasmid pMD2G (Addgene plasmid 12259) using Lipofectamine LTX reagent (Invitrogen) and PLUS REagents following instructions of the supplier (Invitrogen). Next day, the transfection was stopped and 5 mL of medium were added. Two and three days after transfection, lentivirus were collected and stored at −80 ºC. Lentiviral titer was calculated as previously described [62].

## Study of cell viability and growth rate in non-neuronal cells expressing human Tau isoforms

HEK-203T cells were infected by packaged lentiviral particles at an approximated MOI of 5 Infective Units (IU)/cell. Forty-eight and seventy-two hours after infection, photos were taken with the Orca Flash (monochrome sCMOS) (Zeiss) using a 40X dry objective under brightfield illumination. Western blot was performed for infection confirmation. Viability assay was performed following the Calcein AM assay. 2 *µ*M calcein AM working solution in PBS was added to the cells. The plates were incubated at 37 °C for 20 minutes, and calcein AM fluorescence (ex/em 494/517 nm) was read by sCMOS ORCA-Fusion (C11440-20UP) using a 10X/0.3 Aire EC Plan-Neo (Zeiss 420340−9901). The quantification of growth rate and viability was performed using Fiji [63]. Quantitative analysis was done using One-way ANOVA followed by Student's t-test for comparisons.

## Statistical analysis

For quantitative experiments, the statistical significance was determined by Student's t test or One-way ANOVA with Turkey *post-hoc* test, for comparisons including more than two groups. A p value of less than 0.05 was considered significant (* $p < 0.05$; ** $p < 0.01$). Data analysis was conducted using GraphPad Prism (GraphPad Software, La Jolla, CA, USA). For all figures in which error bars are shown, data represent the mean ± SEM. The normality of the data distribution was assessed using the Shapiro-Wilk test.

## Supporting information

**S1 File. Tau 4R2N and Full Tau protein. (A)** Amino acid sequence of Tau 4R2N $(+2, +3, +10)$. **(B)** Amino acid sequence of Full Tau $(+2, +3, +4A, +6, +8)$, an isoform including all exons. Exon 1 starts in blue and alternates with black. Amino acids with a larger font size indicate a codon spanning both exons. Sequences were obtained from ENSEMBL and Uniprotein.
(PDF)

**S2 File. Human Tau proteins expressed in bacteria.** Characterization by gel electrophoresis of Full human Tau and Tau 4R2N isoforms purified from bacteria (see Methods). **(A)** Coomassie brilliant blue staining. **(B)** Western Blot with antibody HT7.
(PDF)

**S3 File. Microtubule binding capacity of Exon 8 Tau isoform.** Electron microscopy images of second polymerization cycle of mouse brain microtubules in the absence (control) or presence of Tau 4R2N or Exon 8 Tau isoform. Circle points thin MTs.
(PDF)

**S4 File. Immuno-MS of amorphous Full Tau aggregates (A)** Immunogold labeling of Full Tau aggregates corresponding to the analysis shown in Figure 3 (white arrows indicate nanogold particles).
(PDF)

**S5 File. Aggregation assays in the absence of heparin. (A)** Aggregation assays of Full Tau and Tau 4R2N carried out without heparin.
(PDF)

**S6 File. Sequence comparison of Exon 8 and Exon 9–10.** Scheme of the comparison between the sequences of Exon 8 and Exons 9–10 (Exon 9-gray, Exon 10-black) where basic amino acids are highlighted inside a box showing their similar location in both sequences.
(PDF)

**S7 File. Raw image dataset supporting the experimental results presented in this study.** This figure includes the unprocessed original images from all key experiments, including Western blots and Coomassie-stained gels.
(PDF)

**S1 Table. Solvent accessible surface areas.** Comparative of Solvent Accessible Surface Areas of both isoforms.
(PDF)

## Author contributions

**Conceptualization:** Laura Vallés-Saiz, Indalo Domene-Serrano, Felix Hernández, Jesús Avila.

**Data curation:** Laura Vallés-Saiz, Indalo Domene-Serrano.

**Formal analysis:** Laura Vallés-Saiz, Indalo Domene-Serrano, Felix Hernández, Jesús Avila.

**Funding acquisition:** Felix Hernández, Jesús Avila.

**Investigation:** Laura Vallés-Saiz, Indalo Domene-Serrano.

**Methodology:** Laura Vallés-Saiz, Indalo Domene-Serrano, Angel J. Picher, Mar Pérez, Vega García-Escudero.

**Writing – original draft:** Laura Vallés-Saiz, Indalo Domene-Serrano, Felix Hernández, Jesús Avila.

**Writing – review & editing:** Laura Vallés-Saiz, Indalo Domene-Serrano, Angel J. Picher, Mar Pérez, Vega García-Escudero, Felix Hernández, Jesús Avila.

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
