## [Decision Letter · Decision Letter 0]

8 Aug 2025

Dear Dr. Avila,

Thank you for submitting your manuscript to PLOS ONE. After careful consideration, we feel that it has merit but does not fully meet PLOS ONE’s publication criteria as it currently stands. Therefore, we invite you to submit a revised version of the manuscript that addresses the points raised during the review process.

We look forward to receiving your revised manuscript.

Kind regards,

Stephen D. Ginsberg, Ph.D.

Section Editor

PLOS ONE

Journal Requirements:

“This work has been supported by grants from the Spanish Ministry of Science: PID2020-113204GB-I00 (F.H.) and PID2021-123859OB-100 from MCIN /AEI/10.13039/501100011033 / FEDER, UE (J.A.). The Centro de Biología Molecular Severo Ochoa (CBMSO) is a Severo Ochoa Center of Excellence (MICIN, award CEX2021-001154-S).”

4. Please note that funding information should not appear in the Acknowledgments section or other areas of your manuscript. We will only publish funding information present in the Funding Statement section of the online submission form. Please remove any funding-related text from the manuscript. 

6. Thank you for stating the following in the Financial Disclosure section: 

“This work has been supported by grants from the Spanish Ministry of Science: PID2020-113204GB-I00 (F.H.) and PID2021-123859OB-100 from MCIN /AEI/10.13039/501100011033 / FEDER, UE (J.A.). The Centro de Biología Molecular Severo Ochoa (CBMSO) is a Severo Ochoa Center of Excellence (MICIN, award CEX2021-001154-S).”

We note that one or more of the authors are employed by a commercial company: Expedeon SLU

2) Please also provide an updated Competing Interests Statement declaring this commercial affiliation along with any other relevant declarations relating to employment, consultancy, patents, products in development, or marketed products, etc.  

Within your Competing Interests Statement, please confirm that this commercial affiliation does not alter your adherence to all PLOS ONE policies on sharing data and materials by including the following statement: ""This does not alter our adherence to  PLOS ONE policies on sharing data and materials.” (as detailed online in our guide for authors http://journals.plos.org/plosone/s/competing-interests) . If this adherence statement is not accurate and  there are restrictions on sharing of data and/or materials, please state these. Please note that we cannot proceed with consideration of your article until this information has been declared.

8. PLOS ONE now requires that authors provide the original uncropped and unadjusted images underlying all blot or gel results reported in a submission’s figures or Supporting Information files. This policy and the journal’s other requirements for blot/gel reporting and figure preparation are described in detail at https://journals.plos.org/plosone/s/figures#loc-blot-and-gel-reporting-requirements and https://journals.plos.org/plosone/s/figures#loc-preparing-figures-from-image-files. When you submit your revised manuscript, please ensure that your figures adhere fully to these guidelines and provide the original underlying images for all blot or gel data reported in your submission. See the following link for instructions on providing the original image data: https://journals.plos.org/plosone/s/figures#loc-original-images-for-blots-and-gels.   

Reviewers' comments:

Reviewer's Responses to Questions

**Comments to the Author**

1. Is the manuscript technically sound, and do the data support the conclusions?

Reviewer #1: Partly

Reviewer #2: Yes

Reviewer #3: Yes

Reviewer #4: Yes

2. Has the statistical analysis been performed appropriately and rigorously?

Reviewer #1: Yes

Reviewer #2: Yes

Reviewer #3: Yes

Reviewer #4: Yes

3. Have the authors made all data underlying the findings in their manuscript fully available?

Reviewer #1: Yes

Reviewer #2: Yes

Reviewer #3: Yes

Reviewer #4: Yes

4. Is the manuscript presented in an intelligible fashion and written in standard English?

Reviewer #1: Yes

Reviewer #2: Yes

Reviewer #3: Yes

Reviewer #4: No

Reviewer #1: The manuscript analyzes, by structural and functional approaches, the isoform of Tau protein named the full tau isoform, containing all exons of the MAPT gene. The paper's content is of interest in the field; however, some implementations are required to be more understandable for the readers and Tau researchers. Furthermore, a key mechanism of action of the Tau protein or Tau isoforms, the phosphorylation of the protein, was not considered, in either the discussion or the experimental design. The material and methods are adequately described, and the results are clear.

Major concern:

The authors have produced the Tau full length utilizing a bacterial model. Since protein phosphorylation is an essential post-translational modification for its function and aggregation mechanism, did the authors verify whether lentiviruses' protein was phosphorylated? Could phosphorylation influence functional aggregation assays? The authors should comment this aspect in the discussion section. It would also be appropriate to verify if the protein is phosphorylated in expression studies on viability and growth.

Authors might declare the limit of the study.

Minor point:

- The denomination tau 42 must be explained by the authors in the manuscript's text.

- In the introduction (lines 66-70), it will be helpful to include at least an additional and recent reference that supports the statement (such as https://doi.org/10.1038/ s41580-022-00545- z or others).

- Lines 64-65, the sentence (Where there are…) is unclear.

- Lines 94-97 check the English style.

- Line 238 "To try to explain the previous, we decide" seems that a world is missing; it might be the world "results".

- To be more informative, specify in figure 1 and the figure caption the differences between panel B (protein bacterial purification) and panel C, western blot using Tau-specific antibody.

In the discussion, authors might declare the limit of the study.

Reviewer #2: In the article entitled “Structure and function of full-length Tau” the authors analyze the full-length Tau isoform by comparing it with Tau 42.

There are a few points that are important to address.

1. Regarding biophysical characterization, aggregation and structure are shown by microscopy, however fluorescence spectroscopy or conformational dynamics data would be missing.

2. Modeling by AlphaFold2 is useful, but the conclusion on domain accessibility is not experimentally validated, such as limited proteolysis assays or FRET.

3. The presence of amorphous aggregates with Full Tau could be characterized by immuno-MS.

4. The amount of Tau in soluble fraction is not shown, which complements the analysis.

5. Regarding Figure 3, the images shown are qualitative, however, they could be improved with a quantitative analysis of the number, length or diameter of the aggregates. A morphological classification could be included, for example fibrillar vs. amorphous.

6. It would be necessary to examine aggregation in the absence of heparin.

7. It is not shown if the aggregates are beta structures.

8. It is not known whether the amorphous Full Tau aggregates are transiently toxic or inert.

9. Regarding figure 4, no quantitative value of surface accessibility or electrostatic maps are provided.

10. A comparative analysis of free energy, intramolecular interactions or contact prediction to support the model is missing.

11. Regarding Figure 5, the use of a single cell line limits the extrapolation to neuronal models. It would be useful to evaluate markers of apoptosis (TUNEL, caspases) to confirm the mechanism of cell death.

Overall, this is a very interesting article that explores the complete Tau isoform, through expression, microscopy, cell functional analysis, modeling and statistical quantification.

Reviewer #3: The manuscript is well written. It is a topic of great importance, and the authors have carefelly studied the role of exons in the structure and function of tau.

1. The grammar needs to be checked throughout the manuscript, as some sentences do not make sense.

2. In introduction, 'Three main Tau transcripts from the MAPT gene have been described", needs elaboration.

3. IN the same paragraph, "none have been found to translate the full-length tau protein" needs more citations and elaboration. It may help the readers if a few sentences on why it has not been possible, can be mentioned.

4. next paragraph of introduction, "essential roles in tau protein" - what kind of roles?

5. Abbreviatons like pLDDT need to be elaborated.

6. A conclusion discussing the significance of the findings may be included, as it will show the author's opinions and future perspectives.

Reviewer #4: The manuscript addresses an interesting topic in the field of neuroscience: the first study of the structure and function of the complete Tau isoform in the brain. While research on the Tau protein has been a topic of study for years, the proposal presented by the authors provides relevant new information that makes the work a valuable contribution to its field. Overall, the manuscript is well-structured and written, allowing for guided reading and supported by robust evidence. The conclusions are appropriate for the study's objective and the scientific argument presented. Regarding the cited literature, fundamental studies and reviews recognized in the field are mentioned. One-sixth of the references are updated; in this case, this could be due to the limited literature on the topic. Minor revisions are suggested:

- The image clarity is low. Increasing the figure resolution is recommended.

- The legend for Figure 1 is incomplete; the abbreviations mentioned in the image do not indicate their meaning.

- Correct the spelling of Control Mts to Control MTs (Figure 2B).

- Add the meaning of the abbreviations used (SN, PLL) to the descriptions of Figures 2C and 3B.

- Correct the spelling of the abbreviation Pll to PLL (Figure 3C).

- The text mentions that the filaments in Figure 3 are marked with a white arrow, but this is not the case.

- Latin words should be written in italics: post hoc.

- Correctly spell the chemical formula MgCl2.

- Write the chemical formula of ammonium acetate.

- Based on the results obtained and the available literature, what do the authors believe could be the substantial function of the complete Tau protein in the brain?

**Do you want your identity to be public for this peer review?** For information about this choice, including consent withdrawal, please see our Privacy Policy

Reviewer #1: No

Reviewer #2: No

Reviewer #3: No

Reviewer #4: **Yes: ** Donaji Chi-Castañeda

---

## [Author Response · Author response to Decision Letter 1]

15 Sep 2025

Dear Prof. Ginsberg,

Editor Plos One.

Please find enclosed the revised manuscript entitled “Structure and function of full-length Tau.”

In this revised version, we have carefully addressed all the points raised by the editor and reviewers, as you indicated.

1. We have ensured that our manuscript and figures comply with PLOS ONE’s style requirements, following the templates provided by the journal.

2. Grant numbers will be double-checked when resubmitting the manuscript.

3. The funders’ role has been added to the financial disclosure section. The amended text now reads:

“This work has been supported by grants from the Spanish Ministry of Science: PID2023-149460NB-I00 (F.H.: Conceptualization, Formal analysis, Funding acquisition, Project Writing – original draft and Writing – review and editing) and PID2021-123859OB-100 from MCIN /AEI/10.13039/501100011033 / FEDER, UE (J.A.: Conceptualization, Formal analysis, Funding acquisition, Project Writing – original draft and Writing – review and editing). 4basebio S.L.U. supported AJP in the form of salary and research materials for the study. The Centro de Biología Molecular Severo Ochoa (CBMSO) is a Severo Ochoa Center of Excellence (MICIN, award CEX2021-001154-S).”

4. Funding information provided in other sections has been removed.

5. Data sharing will be indicated in the submission form. The data are available in the repository: https://doi.org/10.6084/m9.figshare.28351508.v1

6. Roles, responsibilities, and conflicts of interest have been updated to fully adhere to PLOS ONE’s policies and procedures.

7. Supporting information captions have been included at the end of the manuscript.

8. Raw, uncropped, and unadjusted images are included either in the Supporting Information or in the data repository.

Reviewer 1:

Major Concern

The authors have produced the Tau full length utilizing a bacterial model. Since protein phosphorylation is an essential post-translational modification for its function and aggregation mechanism, did the authors verify whether lentiviruses' protein was phosphorylated? Could phosphorylation influence functional aggregation assays? The authors should comment this aspect in the discussion section. It would also be appropriate to verify if the protein is phosphorylated in expression studies on viability and growth.

Full Tau and Tau 4R2N, originate from bacterial cloning and purification. Thus, both proteins lack of post-translational modifications like, phosphorylation, acetylation, glycosylation, … We have been working with unmodified proteins in this first manuscript to focus on those proteins without any modifications. Indeed, many works from other groups had been also performed with unmodified tau with reliable results. In future studies, we will analyze how different types of modifications at specific sites can affect to the function or disfunction of Full Tau isoform. Moreover, the aim of this study is to investigate the function and structure of the full-length Tau protein in the physiological context of the isoform, rather than in the pathological setting where its modifications would indeed occur.

Minor points:

1. The denomination tau 42 must be explained by the authors in the manuscript's text.

The term Tau 42 referred to the tau isoform Tau 4R2N. This has now been changed to the canonical denomination for clarity.

2. In the introduction (lines 66-70), it will be helpful to include at least an additional and recent reference that supports the statement (such as https://doi.org/10.1038/ s41580-022-00545- z or others).

Several references have been added to the statement.

3. Lines 64-65, the sentence (Where there are…) is unclear.

The sentence “Where there are…” has been rewritten for clarification.

4. Lines 94-97 check the English style.

English style has been checked and improved.

5. Line 238 "To try to explain the previous, we decide" seems that a world is missing; it might be the world "results".

Grammar and style throughout the manuscript have been revised and clarified.

6. To be more informative, specify in figure 1 and the figure caption the differences between panel B (protein bacterial purification) and panel C, western blot using Tau-specific antibody.

The caption for Figure 1 has been expanded, including a statement clarifying the origin of both panels.

Reviewer 2:

1. Regarding biophysical characterization, aggregation and structure are shown by microscopy, however fluorescence spectroscopy or conformational dynamics data would be missing.

We don’t consider that such analysis is related to the focus of the present work. Nevertheless, in the near future we will deeply study conformational changes on Full Tau in different environments. Since fluorescence spectroscopy is based on the excitation of the intrinsic fluorescent properties of the molecules analyzed, we do not consider that the results obtained from these assays directly support the objectives of the present project. However, the use of techniques aimed at analyzing conformational changes has indeed been proposed for future studies, which will focus in greater detail on the structural features of the isoform and their post translational modifications.

2. Modeling by AlphaFold2 is useful, but the conclusion on domain accessibility is not experimentally validated, such as limited proteolysis assays or FRET.

The structural modeling obtained through the machine learning program AlphaFold2 illustrates the conformations adopted by the protein depending on the presence or absence of specific exons, and, notably, highlights the conformational differences associated with exon 8. These in silico data provide a rationale for the differences observed in the results presented in Figure 2. In this context, we do not fully see the necessity of including proteolysis assays or FRET analyses, as they are not essential to the main objectives of the present work. Nevertheless, we greatly appreciate the reviewer’s suggestion, and the implementation of these biophysical approaches will indeed be considered in future studies, in which we intend to explore these aspects in greater depth.

3. The presence of amorphous aggregates with Full Tau could be characterized by immuno-MS.

Immuno-MS of amorphous aggregates with Full Tau has been performed and included in supporting information as S4.

4. The amount of Tau in soluble fraction is not shown, which complements the analysis.

Soluble fraction fraction of both isoforms has been included in Figure 2, as suggested by the reviewer.

5. Regarding Figure 3, the images shown are qualitative, however, they could be improved with a quantitative analysis of the number, length or diameter of the aggregates. A morphological classification could be included, for example fibrillar vs. amorphous.

Figure 3 has been improved with several quantitative data to illustrate every detail of the polymers.

6. It would be necessary to examine aggregation in the absence of heparin.

Aggregation of Full Tau and Tau 4R2N, in the absence of heparin, has been included in supporting information as S5.

7. It is not shown if the aggregates are beta structures.

We cannot rule out the presence of β-structures in the aggregates observed in the aggregation assay performed with Full Tau. Indeed, Full Tau shows some of those structures with canonical isoform. When comparing regions of the tau protein such as exons 10 and 11, which are known to form β-structures, we find substantial sequence similarities with exon 8. This resemblance is now shown in S6. Suggesting that exon 8 could also contribute to β-structural arrangements, in addition to exons 10 and 11, within the aggregates.

8. It is not known whether the amorphous Full Tau aggregates are transiently toxic or inert.

These aggregates do not appear to be toxic, as shown in Figure 5, where cell survival following transduction with the Full Tau protein was not reduced compared to the control. This stands in contrast to cells transduced with the 4R2N isoform, where a marked decrease in viability was observed

9. Regarding figure 4, no quantitative value of surface accessibility or electrostatic maps are provided.

The Solvent Accessible Surface Area (SASA) data have been incorporated into the Supporting Information.

10. A comparative analysis of free energy, intramolecular interactions or contact prediction to support the model is missing.

We appreciate the suggestion of techniques that would further enhance and increase the value of the study. However, the primary aim of this work is to provide a functional and structural characterization that introduces the first insights into this isoform. These biophysical and biological approaches will indeed be implemented in future studies, where we will explore these aspects in greater depth.

11. Regarding Figure 5, the use of a single cell line limits the extrapolation to neuronal models. It would be useful to evaluate markers of apoptosis (TUNEL, caspases) to confirm the mechanism of cell death.

As previously mentioned, the aim of this work is to indicate, for the first time, that Full Tau isoform could be expressed and that this tau isoform shows different characteristics to those found for canonical CNS tau isoforms, been Tau 4R2N a good example of those canonical tau isoforms. In this way, cells transduced with Full Tau did not undergo cell death, whereas cells transduced with Tau 4R2N did, consistent with previous reports of Tau 4R2N-induced cytotoxicity. That cell death could occur by Regulate Cell Death (RCD) as apoptosis, pyroptosis, autophagy and ferroptosis… In further studies, we will study those possibilities by the methodology indicated by the reviewer.

We deeply appreciate the indications and comments of the reviewer because after doing some of the experiments suggested, the work has improved thanks to that input, and we anticipate that their ideas will also be useful in future research.

Finally, we thank such a positive final sentence about the work.

Reviewer 3:

1. The grammar needs to be checked throughout the manuscript, as some sentences do not make sense.

Grammar and style throughout the manuscript have been revised and clarified.

2. In introduction, 'Three main Tau transcripts from the MAPT gene have been described", needs elaboration.

The sentence “Three main Tau transcripts…” has been checked and elaborated.

3. In the same paragraph, "none have been found to translate the full-length tau protein" needs more citations and elaboration. It may help the readers if a few sentences on why it has not been possible, can be mentioned.

More citations have been added to support the statement “none have been found…,” and the sentence has been rephrased for clarity.

4. Next paragraph of introduction, "essential roles in tau protein" - what kind of roles?

The roles of tau protein are included with appropriate citations.

5. Abbreviatons like pLDDT need to be elaborated.

Every abbreviation found in the manuscript has been included in the abbreviation section.

6. A conclusion discussing the significance of the findings may be included, as it will show the author's opinions and future perspectives.

The sentences highlighted by the reviewer have been corrected and further explained in greater detail.

Reviewer 4:

1. The image clarity is low. Increasing the figure resolution is recommended.

Figures included in the revised manuscript have been uploaded at 600 dpi resolution to improve clarity.

2. The legend for Figure 1 is incomplete; the abbreviations mentioned in the image do not indicate their meaning.

The caption for Figure 1 has been completed, with the abbreviations been included in the figure and in abbreviations section.

3. Correct the spelling of Control Mts to Control MTs (Figure 2B).

Mts has been checked and changed by MTs.

4. Add the meaning of the abbreviations used (SN, PLL) to the descriptions of Figures 2C and 3B.

Both abbreviations (SN and PLL) have been included in the figure caption and abbreviations section.

5. Correct the spelling of the abbreviation Pll to PLL (Figure 3C).

Sll has been corrected to SLL.

6. The text mentions that the filaments in Figure 3 are marked with a white arrow, but this is not the case.

White arrows have been added to Figure 3.

7. Latin words should be written in italics: post hoc.

Latin words have been formatted in italics.

8. Correctly spell the chemical formula MgCl2 and write the chemical formula of ammonium acetate.

Chemical formulas, both MgCl2 and ammonium acetate (NH4CH3CO2) have been corrected and included.

9. Based on the results obtained and the available literature, what do the authors believe could be the substantial function of the complete Tau protein in the brain?

The amount of Full Tau is much lower than that of CNS canonical isoforms. This difficult the search of its possible function. A possible way to analyze that possible function of Full Tau will be to confirm its presence in any animal model at a specific localization. That data would facilitate the knowledge of its putative function. We will further work on that.

Yours sincerely,

Jesús Avila

---

## [Decision Letter · Decision Letter 1]

25 Sep 2025

Dear Dr. Avila,

Thank you for submitting your manuscript to PLOS ONE. After careful consideration, we feel that it has merit but does not fully meet PLOS ONE’s publication criteria as it currently stands. Therefore, we invite you to submit a revised version of the manuscript that addresses the points raised during the review process.

We look forward to receiving your revised manuscript.

Kind regards,

Stephen D. Ginsberg, Ph.D.

Section Editor

PLOS ONE

Journal Requirements:

Reviewers' comments:

Reviewer's Responses to Questions

**Comments to the Author**

Reviewer #1: All comments have been addressed

Reviewer #2: All comments have been addressed

Reviewer #4: All comments have been addressed

2. Is the manuscript technically sound, and do the data support the conclusions?

Reviewer #1: Yes

Reviewer #2: Yes

Reviewer #4: Yes

3. Has the statistical analysis been performed appropriately and rigorously?

Reviewer #1: Yes

Reviewer #2: Yes

Reviewer #4: Yes

4. Have the authors made all data underlying the findings in their manuscript fully available?

Reviewer #1: Yes

Reviewer #2: Yes

Reviewer #4: Yes

5. Is the manuscript presented in an intelligible fashion and written in standard English?

Reviewer #1: Yes

Reviewer #2: Yes

Reviewer #4: No

Reviewer #1: (No Response)

Reviewer #2: Thank you to the authors for their replies. With the changes made, the manuscript has improved substantially.

Reviewer #4: Standardize the spelling of Tau throughout the manuscript. Sometimes it appears as "tau" and sometimes as "Tau." The same applies to "full tau," which sometimes appears as "Full Tau."

On line 206, the reference 32 is in parentheses, not brackets.

On line 527, subscript the number 2 in the CaCl2 formula.

**Do you want your identity to be public for this peer review?** For information about this choice, including consent withdrawal, please see our Privacy Policy

Reviewer #1: No

Reviewer #2: No

Reviewer #4: No

---

## [Author Response · Author response to Decision Letter 2]

7 Oct 2025

In this revised version, we have carefully addressed all the points raised by the reviewers, as you indicated.

Reviewer 1 and 2:

We thank them for their positive comments.

Reviewer 4:

1. Is the manuscript presented in an intelligible fashion and written in standard English?

The manuscript has been revised to improve grammar and clarity We believe it is now written in standard, intelligible English.

2. Standardize the spelling of Tau throughout the manuscript. Sometimes it appears as "tau" and sometimes as "Tau." The same applies to "full tau," which sometimes appears as "Full Tau."

The spelling of Tau and Full Tau has been standardized throughout the manuscript.

3. On line 206, the reference 32 is in parentheses, not brackets.

The reference has been corrected to [32].

4. On line 527, subscript the number 2 in the CaCl2 formula.

The chemical formula has been corrected to show CaCl₂ with the subscript as indicated.

We also thank the reviewer for these helpful comments.

---

## [Editor Report · Decision Letter 2]

8 Oct 2025

Structure and function of Full-length Tau

PONE-D-25-07400R2

Dear Dr. Avila,

We’re pleased to inform you that your manuscript has been judged scientifically suitable for publication and will be formally accepted for publication once it meets all outstanding technical requirements.

Kind regards,

Stephen D. Ginsberg, Ph.D.

Section Editor

PLOS ONE

---

## [Editor Report · Acceptance letter]

PONE-D-25-07400R2

PLOS ONE

Dear Dr. Avila,

I'm pleased to inform you that your manuscript has been deemed suitable for publication in PLOS ONE. Congratulations! Your manuscript is now being handed over to our production team.

Kind regards,

on behalf of

Dr. Stephen D. Ginsberg

Section Editor

PLOS ONE